# Sequential Diagnostic Approach Using FIB-4 and ELF for Predicting Advanced Fibrosis in Metabolic Dysfunction-Associated Steatotic Liver Disease

**DOI:** 10.3390/diagnostics14222517

**Published:** 2024-11-11

**Authors:** Yeo-Wool Kang, Yang-Hyun Baek, Sang-Yi Moon

**Affiliations:** Department of Internal Medicine, Dong-A University College of Medicine, 32 Daeshingongwonro, Seo-gu, Busan 49021, Republic of Korea; ywkang8756@dau.ac.kr (Y.-W.K.); sang4401@dau.ac.kr (S.-Y.M.)

**Keywords:** enhanced liver fibrosis, liver fibrosis, metabolic dysfunction-associated steatotic liver disease

## Abstract

***Background and Aims***: Multiple non-invasive tests (NITs) for identifying advanced fibrosis in patients with non-alcoholic fatty liver disease (NAFLD) are available, but, due to the limitations of single NITs, the American Association for the Study of Liver Disease (AASLD) guidelines suggest a two-step strategy, combining the Fibrosis-4 Index (FIB-4) score with the Enhanced Liver Fibrosis (ELF) test to improve diagnostic accuracy and minimize unnecessary liver biopsies. However, few real-world studies have used such a sequential approach. We here evaluated the diagnostic accuracy of the ELF test in patients with recently established metabolic dysfunction-associated steatotic liver disease (MASLD) and assessed the clinical utility of applying a two-step strategy, including the ELF test following the FIB-4 score assessment, in patients with MASLD. ***Methods*:** We enrolled 153 patients diagnosed with MASLD who underwent liver biopsy at the Dong-A University Hospital between June 2018 and August 2023. The degree of fibrosis was determined based on liver biopsy results. Various NITs were used, including the Aminotransferase-to-Platelet Ratio Index (APRI), FIB-4 score, NAFLD Fibrosis score (NFS) and ELF test. The diagnostic efficacy of these NITs was evaluated based on the area under the receiver operating characteristic curve (AUROC). Additionally, the performance of each test was further examined both when applied individually and in a two-step approach, where FIB-4 was used followed by ELF testing. Key metrics such as sensitivity, specificity, positive predictive value (PPV), negative predictive value (NPV), and accuracy were used for this analysis. ***Results*:** Overall, 153 patients with MASLD (mean age: 46.62 years; 52.3% men; 28.1% with type 2 diabetes) were included. The performance of the NITs in identifying advanced fibrosis was as follows: the AUROC of the APRI, FIB-4, NFS, and ELF tests were 0.803 (95% confidence interval (CI), 0.713–0.863), 0.769 (95% CI, 0.694–0.833), 0.699 (95% CI, 0.528–0.796), and 0.829 (95% CI, 0.760–0.885), respectively. The combination of the FIB-4 score ≥ 1.30 and the ELF score ≥ 9.8 showed 67.86% sensitivity, 90.40% specificity, a PPV of 75.18%, an NPV of 86.78%, an accuracy of 83.64%, and an AUROC of 0.791 for predicting the diagnosis of advanced fibrosis. This approach excluded 28 patients (71.8%) from unnecessary liver biopsies. ***Conclusions*:** Our study demonstrated that ELF testing maintained diagnostic accuracy in assessing liver fibrosis in patients with MASLD in real-world practice. This test was used as a second step in the evaluation, reducing clinically unnecessary invasive liver biopsies and referrals to tertiary institutions. This approach allows assessment of MASLD severity in primary care settings without requiring additional equipment.

## 1. Introduction

In June 2023, the term “metabolic dysfunction-associated steatotic liver disease” (MASLD) was introduced to replace the concept of non-alcoholic fatty liver disease (NAFLD) to describe liver disease better and potentially reduce stigma [1,2]. The global burden of MASLD is driven by obesity and type 2 diabetes. MASLD currently affects approximately 38% of the adult population worldwide, representing a 50% increase since the 1990s [3,4,5]. Despite the terminological change, assessment of the degree of fibrosis remains crucial for predicting liver-related events and evaluating disease severity [6,7].

Liver fibrosis significantly affects the prognosis of patients with chronic liver disease: advanced fibrosis increases the risk of adverse liver-related outcomes. Currently, liver biopsy is the gold standard for diagnosing metabolic dysfunction-associated steatotic hepatitis and for staging liver fibrosis, but is limited by its invasiveness, risk of complications, and sampling variability [8,9].

Therefore, using appropriate screening methods to identify patients with potentially advanced fibrosis, while avoiding unnecessary biopsies and referrals to tertiary care centers, is important. Non-invasive tests (NITs) for such screening can be broadly categorized into serum tests and imaging evaluations. Imaging evaluations involve various methods using ultrasound (US) and magnetic resonance elastography (MRE) techniques. US elastograpy methods include vibration-controlled transient elastography (VCTE or Fibroscan^®^), point-shear wave elastography (p-SWE), and two-dimensional shear wave elastography (2D-SWE). VCTE is a well-validated method with high diagnostic and prognostic accuracy that is used worldwide [10]. Its limitations include the absence of established cutoff values for diagnosing advanced fibrosis and cirrhosis and the requirement of purchasing equipment [11,12]. Serum markers can be categorized as either indirect or direct. Indirect serum markers, such as the Fibrosis index-4 (FIB-4) score and the NAFLD Fibrosis Score (NFS), are based on routine laboratory parameters combined with demographic factors [13,14]. Direct serum biomarkers related to fibrogenesis, such as the Enhanced Liver Fibrosis (ELF) test, measure three serum biomarkers involved in matrix turnover: tissue inhibitor of metalloproteinase-1, hyaluronic acid, and N-terminal pro-collagen III peptide [15]. While ELF testing is becoming increasingly available, it remains relatively expensive as compared to other indirect serum markers. However, its accuracy is slightly higher than that of the indirect markers [16,17].

In response to these challenges, in 2023, the American Association for the Study of Liver Diseases (AASLD) proposed a two-step approach that considers factors such as diagnostic accuracy, financial considerations, and accessibility [18]. This two-step approach, which combines the FIB-4 score and VCTE or ELF testing has contributed to improving the positive-predictive value (PPV) and specificity for MASLD. Although many studies have described the two-step strategy using the FIB-4 score and VTCE in real-world practice, only a few studies have investigated the two-step screening using the ELF test with the FIB-4 score.

Therefore, this study evaluates whether NITs in newly proposed MASLD patients are inferior or not to those reported in previous NAFLD studies and whether a two-step strategy can reduce unnecessary testing while maintaining the same diagnostic utility in patients with MASLD.

## 2. Materials and Methods

### 2.1. Study Design and Populations

In this retrospective study, we included 153 patients aged 18 or older who had been diagnosed with MASLD and underwent liver biopsy at Dong-A University Hospital between June 2018 and August 2023. The diagnosis of MASLD was confirmed based on ultrasound findings of fatty liver, excluding chronic liver diseases caused by secondary factors such as excessive alcohol intake (weekly alcohol consumption ≥ 210 g in men and ≥140 g in women), drug-induced liver injury, or viral hepatitis. Additionally, the presence of at least one of five metabolic abnormalities was required: overweight or obesity (BMI ≥ 23 kg/m^2^), fasting glucose ≥ 100 mg/dL, blood pressure ≥ 130/85 mmHg or the use of antihypertensive drugs, plasma triglycerides ≥ 150 mg/dL or lipid lowering treatment, and plasma HDL-cholesterol <40 mg/dL for men and <50 mg/dL for women or lipid-lowering treatment. Clinical data, including medical history and laboratory assessments such as complete blood count and standard biochemical tests, were collected within one week of the liver biopsy.

### 2.2. Clinical Measurement and Laboratory Assessment

Clinical data such as age, sex, weight, height, and body mass index (BMI) were recorded, with BMI calculated by dividing weight in kilograms by the square of the height in meters. For the Asian population, obesity is defined as having a BMI of ≥25 kg/m^2^ [19,20]. The patients’ medical records were reviewed to determine if they were receiving treatment for diabetes mellitus (DM). Impaired glucose tolerance (IGT) was classified by fasting blood glucose levels between 100 and 125 mg/dL. Laboratory evaluations were conducted to measure levels of aspartate aminotransferase (AST), alanine aminotransferase (ALT), albumin, platelet count, and HbA1_C_ levels. Normal reference ranges were defined as AST: 0–40 U/L, ALT: 0–41 U/L, platelet count: 150–450 10^9^/L, albumin: 3.4–5.4 g/dL, and HbA1_C_: 5.7–6.4%.

Non-invasive serological panels, such as the AST-to-Platelet Rate Index (APRI), FIB-4 score, NFS, and the ELF score were used to assess fibrosis. These parameters were calculated using the following formulas:

The APRI was calculated using the following formula: [(AST level/upper limit of normal (ULN))/platelet count (10^9^/L)] × 100. (1)

The FIB-4 score was calculated using the following formula: age (years) × AST (U/L)/[platelet count (10^9^/L) × ALT [(U/L)]^(1/2)^]. (2)

The NFS was calculated from age, BMI, impaired fasting glucose or diabetes status, AST/ALT, platelet count, and albumin levels using the following formula: −1.675 + 0.037 × age (years) + 0.094 × BMI (kg/m^2^) +1.13 × IFG or diabetes (yes = 1, no = 0) + 0.99 × AST/ALT ratio − 0.013 × platelet count (×10^9^/L) − 0.66 × albumin (g/dL).(3)

### 2.3. Histology Assessment

All patients underwent US-guided liver biopsy, with tissue samples preserved in formalin and embedded in paraffin. At our institution, we use a 16-gauge needle for liver biopsies, aiming to obtain tissue samples approximately 1–2 cm in length. Two micrometer-thick sections were stained with hematoxylin, eosin, and Masson’s trichrome. An experienced liver pathologist at Dong-A University examined the biopsy specimens. The liver fibrosis stage was evaluated according to the system developed by the Pathology Committee of the Non-Alcoholic Steatohepatitis Clinical Research Network (CRN) [21]. Fibrosis stages were classified as follows: F0, no fibrosis; F1, mild to moderate zone 3 perisinusoidal or periportal fibrosis; F2, zone 3 perisinusoidal and periportal fibrosis; F3, bridging fibrosis; and F4, cirrhosis. Significant fibrosis was defined as scores F2–F4, while advanced fibrosis was identified by scores F3–F4.

### 2.4. Statistical Analysis

Statistical analyses were conducted using IBM SPSS Statistics (version 22.0; IBM Corp., Armonk, NY, USA) and MedCalc Software for Windows version 17.1 (MedCalc Software Ltd., 2022, Ostend, Belgium). Continuous variables were presented as means with standard deviations, while categorical variables were expressed as number with percentages (%). To evaluate the diagnostic performance of APRI, FIB-4, NFS, and ELF, the area under the receiver operating characteristic curve (AUROC) along with 95% confidence intervals (CI) was calculated. The optimal cutoff for each test was determined using the Youden index. The AUROC values for APRI, FIB-4, NFS, and ELF in detecting advanced fibrosis were compared using the DeLong test. A *p*-value of less than 0.05 was considered statistically significant.

### 2.5. Ethics Committee Approval

Ethics approval was granted by the Dong-A Medical School Ethics and Medical Research Committee (DAUHIRB-17-197). Prior to enrollment, written informed consent was obtained from each patient.

## 3. Results

### 3.1. Baseline Characteristics of the Study Population with and Without Advanced Fibrosis

The baseline characteristics of the study population (*n* = 153) and the comparison groups according to the presence of advanced fibrosis (F3–F4) are summarized in Table 1. The mean age of the patients was 46.62 years, with 80 (52.3%) being male. Metabolic conditions observed included obesity (BMI ≥ 25 kg/m^2^) in 95 patients (62.09%), diabetes mellitus (DM) in 43 patients (28.1%), and impaired glucose tolerance (IGT) in 67 patients (43.79%). Liver biopsy results showed fibrosis stages of F0 in 69 patients (45.1%), F1 in 44 patients (28.8%), F2 in 12 patients (7.8%), F3 in 26 patients (17.0%), and F4 in 2 patients (1.3%). When comparing groups according to the presence or absence of advanced fibrosis, the group with advanced fibrosis was older and had a higher prevalence of DM and hypertension that did the group without advanced fibrosis. However, metabolic factors, such as blood pressure, diabetes, obesity, and dyslipidemia, were not significantly different between the two groups, and only AST levels were significantly higher in the advanced fibrosis group than in the non-advanced fibrosis group.

### 3.2. Comparing the Performance of Each Noninvasive Test in Detecting Advanced Fibrosis

The AUROCs of APRI, FIB-4, NFS, and ELF for advanced fibrosis (F3–F4) were 0.803 (95% confidence interval [CI], 0.731–0.863), 0.769 (95% CI, 0.694–0.833), 0.699 (95% CI, 0.528–0.796), and 0.829 (95% CI, 0.760–0.885), respectively (Figure 1). When we compared the performance of each NIT, the APRI, FIB-4, NFS, and ELF demonstrated considerable diagnostic utility in detecting advanced fibrosis, with acceptable AUROC values across the board. However, among these, the ELF test showed relatively superior diagnostic performance, with an AUROC of 0.829 (95% CI, 0.760–0.885), which was slightly higher than those of the APRI, FIB-4, and NFS scores, highlighting its superiority in detecting advanced liver fibrosis. With a cutoff value of >9.6, the ELF test achieved a sensitivity of 67.86%, a specificity of 84.00%, a PPV of 48.7%, and a negative-predictive value (NPV) of 89.2%. The overall diagnostic accuracy was 81.1%, with a *p*-value of <0.0001, indicating strong statistical significance. Although the APRI, FIB-4, and NFS score also performed well, the ELF test provided a more balanced combination of sensitivity and specificity. This suggests that, although all four NITs are valuable in clinical practice, the ELF test may offer relatively better ability to detect advanced fibrosis, making it a particularly useful tool for primary clinicians. Additionally, we analyzed serum ELF and other serum markers according to the fibrosis stage. Spearman’s rank correlation analysis showed that the APRI (Spearman’s r = 0.456, *p* = 0.000), FIB-4 (Spearman’s r = 0.311, *p* = 0.000), NFS (Spearman’s r = 0.267, *p* = 0.001), and ELF (Spearman’s r = 0.410, *p* = 0.000) scores increased as fibrosis severity progressed. The correlation coefficients assess the strength and direction of the association between serum markers and fibrosis stage.

### 3.3. Two-Step Strategy for a Diagnostic Algorithm Combining FIB-4 and Serum ELF for Advanced Fibrosis

Based on the AASLD guidelines for discriminating advanced fibrosis, we first employed the FIB-4. To those with a FIB-4 score between 1.30 and 2.67, who were an indeterminate group, the second step ELF classification was subsequently applied. When we applied the first step, i.e., FIB-4, the indeterminate group accounted for 27.5% (*n* = 42) of all participants. We then calculated the sensitivity, specificity, PPV, and NPV of our proposed algorithm, that involved applying an ELF score cutoff of 9.8 to the indeterminate FIB-4 group and found a sensitivity of 67.85%, a specificity of 90.40%, a PPV of 75.18%, and an NPV of 86.77% (Figure 2). By applying this cutoff for the ELF score in the indeterminate group, unnecessary testing could be avoided in 28 of 42 patients (66.7%).

### 3.4. Propose Three-Step Diagnostic Algorithm Combining the FIB-4, ELF Score, and an Additional Imaging Test for Advanced Fibrosis

We further tested the performance of a three-step diagnostic algorithm, combining the FIB-4, ELF, and additional imaging for detecting advanced fibrosis. The performance of the three-step strategy is shown in Figure 3. The three-step strategy adding the acoustic radiation force impulse (ARFI) to the FIB-4 and ELF tests had a sensitivity of 75.00%, a specificity of 90.40%, a PPV of 77.00%, and an NPV of 89.40%. Compared to the two-step approach, the accuracy and AUROC values were higher, at 85.78% and 0.827, respectively, indicating that the three-step approach improved diagnostic performance.

## 4. Discussion

In this study, our primary objective was to establish simple and reliable criteria for identifying advanced fibrosis in patients diagnosed with MASLD using NITs. Advanced fibrosis, defined as stages F3–F4, represents a critical threshold for liver disease progression in which the risks of both all-cause and liver-related mortality increases substantially [22]. Early identification of fibrosis is essential as it allows for timely and potentially life-saving interventions that can halt or slow disease progression.

Multiple NITs have been extensively validated and adopted to identify advanced fibrosis in patients with MASLD, formerly known as NAFLD [23]. We revealed that various NITs demonstrated similar overall predictive values for detecting advanced fibrosis in patients with MASLD and NAFLD, implying that existing diagnostic tools can effectively be applied to this newly categorized disease.

Several NITs have been developed as alternatives to liver biopsy, which is currently the gold standard for the assessment of liver fibrosis. Among these, the APRI, FIB-4, and NFS are notable for their reliance on basic clinical and laboratory data, which makes them accessible and practical tools for clinicians. NITs have been widely used in numerous studies as screening tools to detect advanced fibrosis [24,25]. Nevertheless, despite their utility, these tests have limitations. For instance, while useful, the APRI has decreased accuracy in detecting early stages of fibrosis and yields a high proportion, approximately 30%, of indeterminate results [26,27]. This limitation can lead to diagnostic uncertainty and may necessitate additional testing or even liver biopsy, which can be invasive and costly.

Similarly, the FIB-4 and NFS, which are the most extensively validated tests for NAFLD, produce scores in the “indeterminate” range in at least 30% of cases [26]. This indeterminacy is particularly pronounced in specific age groups [28]. For example, studies have shown that the FIB-4 and NFS have reduced specificity in patients aged 65 years and older, leading to the proposal of new cutoff values tailored for this age group. Conversely, in younger patients, specifically those under 35 years of age, these tests have demonstrated decreased accuracy, prompting the need for alternative assessments [29].

VCTE is a widely used and effective tool for liver stiffness measurements, but may not be feasible for use in 5–13% of cases, particularly due to obesity or other technical limitations that affect the accuracy of the readings [12,30]. These limitations present challenges, particularly in primary care centers where advanced imaging equipment may not be readily available. Moreover, MRE tests are highly accurate but expensive, limiting their use in all patients in the clinic, and hampering improvement tracking.

Considering these limitations, recent clinical studies have recommended the development of a two-step diagnostic strategy to improve the accuracy of diagnosing advanced fibrosis in patients with NAFLD. A prospective longitudinal cohort study involving 3012 individuals employed a two-step strategy comprising FIB-4, followed by ELF assessment [31]. The study reported that unnecessary referrals were reduced by 80%, and that the diagnosis of advanced fibrosis and cirrhosis was improved 5-fold and 3-fold, respectively as compared with diagnosis using traditional approaches. These findings underscore the potential of the two-step strategy not only to enhance diagnostic precision but also to optimize resource utilization in healthcare settings.

In our study, the two-step approach using FIB-4 and the ELF score demonstrated a sensitivity of 67.85%, a specificity of 90.40%, a PPV of 75.18%, and an NPV of 86.77%. Additionally, it significantly reduced the need for unnecessary liver biopsies. One of the most significant contributions of our study is the introduction of a two-step diagnostic strategy that combines two NITs to enhance predictive accuracy and minimize the need for unnecessary liver biopsies [18]. The proposed two-step strategy begins with the FIB-4 as the first step in excluding patients who are unlikely to have advanced fibrosis. This is followed by a secondary test, such as VCTE or ELF, to improve the diagnostic accuracy and reduce the number of patients classified as indeterminate.

In our study, we opted to use the ELF assessment as part of the two-step strategy. Unlike VCTE, ELF is less affected by factors such as obesity and abdominal fat depth, which can compromise the accuracy of VCTE readings [32]. Furthermore, ELF provides an objective assessment of liver fibrosis, making it a more feasible option for widespread use, particularly in settings where access to specialized equipment is limited [33,34].

Although ELF tests are widely used in Europe, research focusing on Asian populations, particularly in the context of the MASLD, has been limited. Our findings confirmed that the ELF assessment maintained its diagnostic accuracy in patients with MASLD, supporting its use as a secondary diagnostic tool following FIB-4 in the two-step strategy. The application of this sequential strategy in patients with MASLD not only demonstrated reliable diagnostic capabilities but also reduced the need for unnecessary liver biopsies and tertiary care referrals, consistent with findings from other studies.

Additionally, our study explored the potential of ARFI elastography as a third step test, a technique performed concurrently with US, as an adjunctive tool in cases where FIB-4 and ELF results were in the indeterminate range of 7.7 to 9.8 [35]. The inclusion of the ARFI showed a slight improvement in diagnostic predictive power. However, from a cost-effectiveness perspective, the ARFI did not demonstrate outstanding superiority. The values presented by the ARFI vary between devices, and the lack of established common cut-off values limits its use as a definitive diagnostic guide [36,37].

This study had some limitations. First, selection bias may have been present, because the research was conducted at a single institution, and only patients who underwent liver biopsy were included. Therefore, the results may not be generalizable to a broader population. Future research should involve larger multicenter cohorts to validate these findings and ensure broader applicability. Second, VCTE is the most widely used and extensively studied tool to assess liver fibrosis. However, VCTE was not performed in the present study. This omission represents a limitation as it precludes a direct comparison between using ELF as the second step in the two-step strategy and using TE in that role. Analyzing the efficacy of applying ELF versus TE in the second step could have provided valuable insights into the relative strengths and limitations of each method in the diagnostic process. Without this comparison, the findings of the two-step strategy may have been incomplete.

In conclusion, the ELF score is a simple yet effective serum-based test with sufficient diagnostic ability for detecting advanced fibrosis, making it suitable for implementation in primary care settings for patients with MASLD. The two-step strategy that applies FIB-4 followed by the ELF test offers improved diagnostic performance in real-world settings. This approach is not only simpler and more practical, but also enhances its suitability for primary care centers. Moreover, this strategy aligns with the broader goal of reducing the burden on healthcare systems by minimizing unnecessary procedures and focusing resources on patients with the highest risk of disease progression. As MASLD continues to be recognized as a significant public health concern, the adoption of streamlined and effective diagnostic strategies is crucial for improving patient outcomes and optimizing healthcare delivery.

## Figures and Tables

**Figure 1 diagnostics-14-02517-f001:**
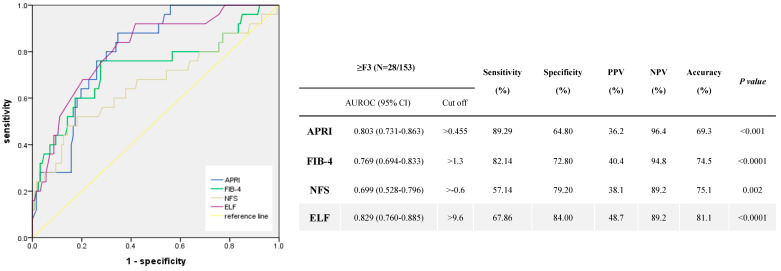
Diagnostic performance for advanced fibrosis (≥F3) in at risk patients in MASLD.

**Figure 2 diagnostics-14-02517-f002:**
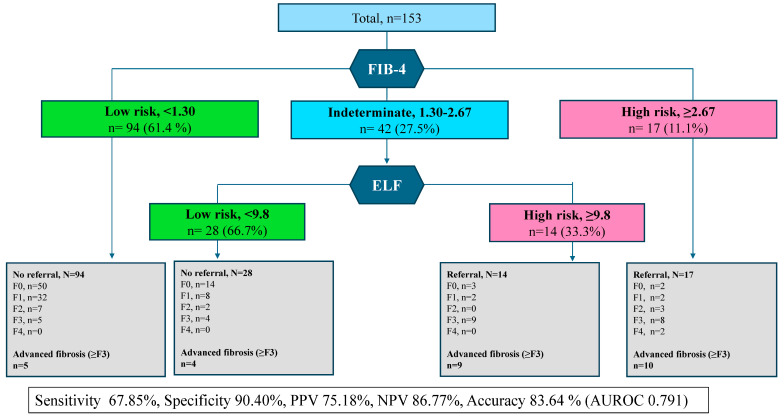
Real-world two-step strategy for identify advanced fibrosis with MASLD.

**Figure 3 diagnostics-14-02517-f003:**
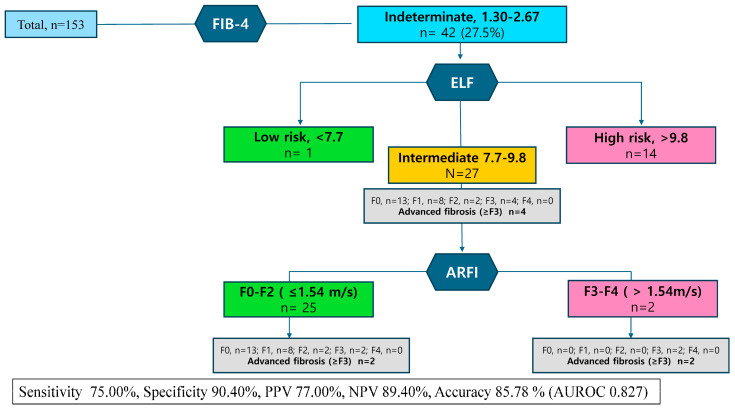
Three-step strategy for identifying fibrosis with indeterminate MASLD patients.

**Table 1 diagnostics-14-02517-t001:** Clinical and demographic characteristics of the patients.

		Advanced Fibrosis	No Advanced Fibrosis	*p*-Value	All
*N*		28	125		153
Age		50.75 ± 15.92	45.70 ± 13.66	0.0886	46.62 ± 14.18
Sex	Male	10 (39.3%)	70 (56.0%)	0.5207	80
	Female	18 (60.7%)	55 (44.0%)		73
Body mass index		30.15 ± 3.96	28.66 ± 4.21	0.0892	28.93 ± 4.20
Type 2 diabetes		11 (39.3%)	32 (25.6%)	0.1453	43
Hyperlipidemia		15 (53.6%)	76 (60.8%)	0.4812	91
Hypertension		18 (64.3%)	63 (50.4%)	0.1833	81
AST, U/L		86.29 ± 50.96	48.47 ± 36.33	0.0001	55.39 ± 41.87
ALT, U/L		93.50 ± 91.73	66.39 ± 66.98	0.0738	71.35 ± 72.57
Platelet count, 10^9^/L		243.64 ± 81.58	272.34 ± 71.10	0.0623	267.09 ± 73.69
FIB-4 score		2.43 ± 1.85	1.13 ± 0.70	0.0001	1.37 ± 1.12
ELF score		10.11 ± 1.15	8.70 ± 1.00	0.0001	8.96 ± 1.17
APRI		0.94 ± 0.60	0.47 ± 0.36	0.0001	0.55 ± 0.45
NFS		−0.80 ± 2.31	−2.09 ± 1.70	0.0009	−1.85 ± 1.89
Fibrosis					
F0 (*n*, %)					69, 45.1%
F1 (*n*, %)					44, 28.8%
F2 (*n*, %)					12, 7.8%
F3 (*n*, %)					26, 17.0%
F4 (*n*, %)					2, 1.3%

## Data Availability

The data presented in this study are available on request from the corresponding author. The data are not publicly available due to including some personal information.

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
