# Peer review of "Sequential Diagnostic Approach Using FIB-4 and ELF for Predicting Advanced Fibrosis in Metabolic Dysfunction-Associated Steatotic Liver Disease"

_diagnostics, 2024, doi:10.3390/diagnostics14222517_

Round 1

Reviewer 1 Report

Comments and Suggestions for Authors

This study assessed the diagnostic accuracy of several noninvasive tests (APRI, FIB-4, NFS, ELF) for prediction of advanced fibrosis (F >=3) on liver histology. In addition, the performance of sequential strategies (2-step: FIB-4->ELF, 3-step: FIB-4->ELF->ARFI) are evaluated.

The mythology is sound and the paper is well written.

However, some points require further attention:

1. Interestingly APRI shows higher AUROC than FIB-4. Therefore it would be interesting to see the diagnostic performance of a sequential 2-step strategy using APRI followed by ELF test.

2. The quality of the reference standard, liver biopsy, should be described in more detail:

a. Which type of biopsy needles were used? Which diameter?

b. What were the criteria for a representative biopsy specimen? The length of the biopsy cylinders and number of portal tracts should be reported.

Author Response

Comments1. Interestingly APRI shows higher AUROC than FIB-4. Therefore it would be interesting to see the diagnostic performance of a sequential 2-step strategy using APRI followed by ELF test.

Response 1) Thank you for pointing this out. I agree with this comment. But, The purpose of this study was to demonstrate the clinical utility of the two-step diagnostic approach recommended by the AASLD and to examine the role of the ELF test as a second-step method. Therefore, we used FIB-4 as the initial screening method, as suggested in the guidelines.

Comments2. 

The quality of the reference standard, liver biopsy, should be described in more detail:

  1. Which type of biopsy needles were used? Which diameter?
  2. What were the criteria for a representative biopsy specimen? The length of the biopsy cylinders and number of portal tracts should be reported.

Response 2 ) I will add this information to the manuscript (highlighted in yellow in the main text).

At our institution, 16-gauge needles were used to obtain biopsy specimens measuring 1 to 2 cm.

Reviewer 2 Report

Comments and Suggestions for Authors

In this manuscript titled “Sequential Diagnostic Approach by Using FIB-4 and ELF for Predicting Advanced Fibrosis in Metabolic Dysfunction-Associated Steatotic Liver Disease”, Kang and colleagues evaluated the diagnostic accuracy of the Enhanced Liver Fibrosis (ELF) test in patients with metabolic dysfunction-associated steatotic liver disease (MASLD) and assessed the clinical utility of applying a two-step strategy, including the ELF following the FIB-4 score assessment. Here are my comments and suggestions related to this manuscript.

-In the conclusions section related to the abstract, why is FIB-4 not considered?

-The authors should clarify that when the liver shows fibrosis the disease is referred to as Metabolic dysfunction-associated steatohepatitis (MASH).

-In the introduction section, please describe the laboratory parameters and demographic factors, which are considered for fibrosis index-4 (FIB-4).

-Clarify if the patients were not infected with COVID-19. 

-The materials and methods section should be improved, please include what tests were applied to exclude liver diseases

-Why were medications taken by patients for comorbidities not included? 

-Histologic stains used in this study are missing, please add this information in materials and methods.

-It is not clear how the criteria for MASLD were assessed?

Author Response

Thank you for your thoughtful advise and comments.

Comments1. In the conclusions section related to the abstract, why is FIB-4 not considered?

Response 1) Numerous studies have already assessed the diagnostic accuracy of non-invasive serum markers, including FIB-4, APRI, and NFS, which are also presented in this study. The AASLD recently recommended a two-step approach: FIB-4 as the initial screening step, followed by the ELF test or transient elastography as the second step. This study’s purpose is to emphasize the utility of non-invasive testing in a sequential approach, as suggested by recent AASLD guidelines, and thus highlights the value of ELF in the second step.

Comments 2. The authors should clarify that when the liver shows fibrosis, the disease is referred to as Metabolic dysfunction-associated steatohepatitis (MASH).

Response 2) In the MASLD disease continuum, MASH typically indicates the presence of inflammation without fibrosis progression. This study is not designed to evaluate the degree of inflammation, and therefore, explanations or mentions of MASH have been minimized.

Comments 3. In the introduction section, please describe the laboratory parameters and demographic factors considered for fibrosis index-4 (FIB-4).

Respones 3) Among liver disease researchers, FIB-4 is a widely recognized formula, and the parameters used to calculate each non-invasive test, including FIB-4, are specified in the Methods section.

Comments 4. Clarify if the patients were not infected with COVID-19.

Response 4) Since there is no well-defined link between COVID-19 and MASLD, COVID-19 status was not included as an essential criterion for patient inclusion. However, we recognize this as a potential area of consideration for future studies.

Comments 5. The materials and methods section should be improved; please include what tests were applied to exclude liver diseases.

Respons 5) In the "Study Design and Population" subsection of the Methods, we described the exclusion of patients with other liver diseases, which were determined based on hematological tests.

Comments 6. Why were medications taken by patients for comorbidities not included?

Response 6 ) Currently, there are no medications that are conclusively known to impact treatment or prognosis in fatty liver disease, so specific medications were not included or used as criteria for patient exclusion.

Comments 7. Histologic stains used in this study are missing; please add this information in materials and methods.

Response 7 ) We will add this information to the manuscript (highlighted in green in the main text).

Comments 8. It is not clear how the criteria for MASLD were assessed.

Response 8 ) We have clarified the MASLD criteria in the "Study Design and Population" section of the Methods (highlighted in green in the main text). This part is very helpful advice in making the content of your research clearer. thank you for your advise.

Round 2

Reviewer 1 Report

Comments and Suggestions for Authors

All points have been answered satisfactorily.

Author Response

All points have been answered satisfactorily.

A ) Thank you for your thoughtful review

Reviewer 2 Report

Comments and Suggestions for Authors

Here are my comments and suggestions related to the second version of this manuscript. 

-The authors should clarify in the materials and methods section that in this study it does not evaluate the degree of inflammation. 

Author Response

Comments 1 : The authors should clarify in the materials and methods section that in this study it does not evaluate the degree of inflammation. 

Response 1 : Thank you for your thoughtful comments. In liver disease, the key factor influencing prognosis and severity is the degree of fibrosis. Therefore, this study aimed to identify effective methods for predicting fibrosis in patients with liver disease. The standard method for assessing fibrosis is a liver biopsy, during which the NASH CRH system is applied. Consequently, even if the degree of inflammation is not evaluated, this assessment system is still utilized. Since this study did not address inflammation, we have removed the section describing criteria for evaluating it. Thank you again.